# Characteristics of Multiple Acute Concomitant Cerebral Infarcts Involving Different Arterial Territories

**DOI:** 10.3390/jcm12123973

**Published:** 2023-06-11

**Authors:** Naaem Simaan, Leen Fahoum, Andrei Filioglo, Shorooq Aladdin, Karine Wiegler Beiruti, Asaf Honig, Ronen Leker

**Affiliations:** 1Department of Neurology, Ziv Medical Center, Safed 13100, Israel; naaems@ziv.gov.il; 2Azrieli Faculty of Medicine, Bar Ilan University, Safed 52900, Israel; 3Department of Neurology, Hadassah-Hebrew University Medical Center, Jerusalem 91120, Israel; leen.fahoum@mail.huji.ac.il (L.F.); afilioglo@gmail.com (A.F.); shorooqaladdin@gmail.com (S.A.);; 4Research Wing, Ziv Medical Center, Safed 13100, Israel; karinebeiruti@ziv.gov.il

**Keywords:** cerebral infarcts, acute single embolic stroke, shower of emboli, arterial territories, encephalopathy

## Abstract

(1) Background: Multiple acute concomitant cerebral infarcts (MACCI) are relatively uncommon. Data regarding the characteristics and outcomes of patients with MACCI are lacking. We, therefore, aimed to characterize the clinical features of MACCI. (2) Methods: Patients with MACCI were identified from a prospective registry of stroke patients admitted to a tertiary teaching center. Patients with an acute single embolic stroke (ASES) involving only one vascular bed served as controls. (3) Results: MACCI was diagnosed in 103 patients who were compared to 150 patients with ASES. MACCI patients were significantly older (*p* = 0.010), more often had a history of diabetes (*p* = 0.011) and had lower rates of ischemic heart disease (*p* = 0.022). On admission, MACCI patients had significantly higher rates of focal signs (*p* < 0.001), an altered mental state (*p* < 0.001) and seizures (*p* = 0.036). The favorable functional outcome was significantly less common in patients with MACCI (*p* = 0.006). In the multivariable analysis, MACCI was associated with lower chances of achieving favorable outcomes (odds ratio: 0.190, 95% CI: 0.070–0.502). (4) Conclusions: There are important differences in clinical presentation, comorbidities and outcomes between MACCI and ASES. MACCI is less often associated with favorable outcomes and could represent a more severe form of a stroke compared with a single embolic stroke.

## 1. Introduction

The clinical presentations of an ischemic stroke vary considerably depending on the affected brain regions. The majority of ischemic strokes involve only a single vascular territory. Multiple acute concomitant cerebral infarcts involving multiple arterial territories (MACCI) are also known as “showers of emboli” and account for almost 10% of all ischemic strokes [1,2,3,4]. Due to the simultaneous involvement of several arterial territories, the clinical manifestations of MACCI may have distinct clinical features [2].

In the last decade, there has been an increasing use of advanced brain imaging techniques, such as diffusion-weighted brain MRI (DWI) [5,6], to diagnose the affected vascular region in stroke patients. Compared to computed tomography (CT) or conventional MRI sequences such as T1 or T2 weighted images, DWI can more readily identify acute and hyperacute infarcts with greater sensitivity and specificity, to visualize lacunar infarcts and differentiate recent infarcts from old ones [7,8]. This imaging technique enhanced the detection of MACCI in clinical practice [2,4].

The main etiologies that can cause MACCI include a brain embolism from the heart or the aortic arch, hypo-perfusion, hyper-viscosity and coagulopathies [4,7,9,10]. Although the main etiology has been thought to be embolic in origin, arising either from the heart or large vessels such as the aorta and carotid arteries, several other studies [11,12] have shown the role of non-embolic causes such as hematological disorders and malignancy [13], which challenges secondary prevention treatment strategies in these patients. Therefore, these patients may require a focused and frequent assessment of tissue perfusion with the dosage of blood lactate [14], capillary refill time [15] and blood coagulation monitoring using rotational thromboelastometry and thromboelastography [16]. Consequently, patients with MACCI need adequate anticoagulation therapy [17]. However, the clinical attributes and outcomes in patients with MACCI are not fully understood. Therefore, we aimed to compare the clinical features, comorbidities, stroke characteristics and outcomes between patients with MACCI and those with a stroke that only involved a single vascular territory.

## 2. Materials and Methods

Consecutive patients with acute ischemic stroke were recruited into an ongoing institutional database from Hadassah-Hebrew University Medical Center in Jerusalem, Israel. The institutional review board approved the study with an exemption from obtaining individual informed consent forms due to the anonymized nature of data collection and the retrospective analysis of data (0713-20-HMO).

In the current study, we included all patients diagnosed with MACCI between January 2015 and December 2020. All included patients had a MRI stroke study that included diffusion-weighted imaging (DWI) at b0, b500 and b1000, adjusted diffusion coefficient (ADC) and fluid-attenuated inversion recovery (FLAIR) sequences. Studies were performed on 1.5T GE (Voyager), 3T Philips (Ingenia) or GE (Premier) MRI machines. MRI studies were read blindly for the discharge diagnosis by experienced neuroradiologists and stroke neurologists. Patients who did not have MRI (e.g., metal shrapnel, pacemakers, etc.) were excluded. In addition, included patients also underwent CT angiography unless there were contraindications to iodine administration such as known allergies to iodine or progressive renal impairment. MACCI was defined as multiple acute cerebral infarcts on DWI MRI involving more than one individual vascular territory that could not be attributed to a single parent arterial territory. In this context, infarctions involving the right anterior cerebral artery (ACA) and right middle cerebral artery (MCA) would not qualify as MACCI because both could be attributed to the right internal carotid. Additionally, infarctions involving the right middle and posterior cerebral arteries would not qualify as MACCI if there was an embryonic posterior cerebral pattern because both could be attributed to the right internal carotid. In contrast, patients with acute infarctions involving the left and right anterior circulation or the anterior and any posterior circulation territorial infarctions, in the absence of an embryonic posterior cerebral origin, would qualify as having MACCI.

Baseline characteristics and comorbidities including the presence of diabetes, hyperlipidemia, smoking, hypertension, atrial fibrillation, prior stroke and other possible pre-morbid factors were documented.

Data regarding the clinical manifestations such as the neurological deficits assessed with the National Institutes of Health Stroke Scale (NIHSS) [18,19] on admission, on day 1 and on discharge were obtained from the medical records. Seizure occurrence and type as well as uncommon features upon presentation such as encephalopathy, delirium or acute cognitive impairment were also documented. Cognitive impairment was defined as any acute memory, language or judgment decline that interferes with patient’s daily activity. Encephalopathy was defined as any acute alteration of mental state. Delirium, which frequently occurs in stroke patients, especially in the ICU setting [20], was characterized by impairment in attention, orientation, sleep cycle and consciousness [21].

Follow-up examinations were performed at the stroke outpatient clinics and functional outcomes were assessed with the modified Rankin Scale (mRS) score at 90 days post-stroke. Consecutive patients with acute single embolic stroke (ASES) who were included in the same prospective institutional database during the same period and who completed inpatient MRI and had complete outpatient follow-up examinations were included as controls. We specifically chose patients with ASES as controls because we speculated that they will closely resemble those with MACCI in terms of stroke pathogenesis. For example, cardio-embolic strokes may lead to single or multiple ischemic lesions in a single vascular territory (ASES) or scattered and bilateral ones involving more than one vessel territory which is defined as multiple acute concomitant cerebral infarcts involving multiple arterial territories (MACCI), with the only presumed difference being the number of involved territories. Figure 1 exhibits examples of MRI-DWI results in patients diagnosed with ASES (Figure 1A) or MACCI (Figure 1B).

Patients were evaluated according to the standing standard institutional operational protocol and admission to intensive care units was as per the attending physicians’ discretion in accordance with institutional guidelines that were based on AHA guidelines [22]. We acquired and compared data regarding patient demographics, clinical features, comorbidities, stroke characteristics and outcome and possible etiologies. Neurological deficits upon admission and discharge were measured with the National Institutes of Health Stroke Scale (NIHSS) [18,19]. Additional parameters included survival and favorable outcome rates defined as a modified Rankin scale (mRS) (score of 0–2 for patients without prior disability and no change in mRS for those who had a pre-stroke mRS of >2) [23]. Patients were followed up at the stroke outpatient clinic between 3 and 6 months after the index event. Factors associated with outcomes were studied in the entire cohort.

All patients underwent a detailed series of investigations and etiological workup. Additionally, 12-lead electrocardiogram (ECG), transthoracic echocardiography, cardiac telemetry monitoring for at least 48 h and imaging of both extra- and intracranial arteries with cervical duplex and either CT angiography or MR angiography were used. Patients also underwent an extensive laboratory workup including complete blood count, routine blood biochemistry, protein C, protein S, antithrombin III, factor V Leiden, factor VIII, prothrombin, homocysteine, anti-cardiolipin antibodies, anti-beta-2-glycoprotein antibodies and lupus anticoagulant. Patients with MACCI or ASES with unknown etiology underwent further workup in order to rule out occult malignancy including chest, abdomen and pelvis CT scans and a battery of neoplastic serum markers such as CEA and CA-125, as well as clinical evaluation of the skin by a dermatologist to rule out melanoma. In cases of highly suspected malignancy, females underwent mammography or ultrasound to rule out breast cancer while males underwent clinical examination by a urologist to rule out testicular tumors.

Statistical evaluation: Data analysis was performed with SPSS version 24 (IBM Corp. Released 2016. IBM SPSS Statistics for Windows, Version 24.0. Armonk, NY: IBM Corp USA). The two-sample *t*-test was applied for testing differences between the study groups for quantitative parameters. Pearson’s chi-squared or Fisher’s exact tests were applied for testing the differences between the groups for categorical parameters. *p*-value of 5% or less was considered statistically significant. Multivariate unconditional logistic regression models controlling for age, stroke severity, presentation variables and risk factors that yielded a *p*-value of <0.1 in univariate analysis, as well as the presence of MACCI, were used to assess outcome measures including survival and favorable functional outcome. Data are presented as odds ratios (OR) and 95% confidence intervals (95% CI).

## 3. Results

Out of 4683 stroke admissions between 2015 and 2020, 103 patients with MACCI were identified (2.1% of all stroke cases) and compared to 150 random consecutive patients with ASES from the same database. Data were retrieved from electronic medical records including the index event as well as follow-up data from the stroke outpatient clinic.

Baseline characteristics of the two groups are shown in Table 1. Patients with MACCI more often had diabetes (53% vs. 37%, *p* = 0.011), but less often suffered from ischemic heart disease (26% vs. 44%, *p* = 0.004). Other common vascular risk factors and use of stroke prevention therapies at baseline did not significantly differ in frequency between the groups (Table 1).

Upon presentation, MACCI patients had significantly higher rates of focal neurological signs such as hemiparesis, dysphasia or hemihypoesthesia (74% vs. 39%, *p* < 0.001), and more often presented with an altered mental state (27% vs. 6%, *p* < 0.001) and seizures (18% vs. 9%, *p* = 0.036). However, the initial stroke severity was similar and a similar percentage of patients were treated with systemic thrombolytics in both groups. The rates of ICU admission were higher for ASES patients (45% vs. 23%, *p* < 0.001). The frequency of the epileptic status or use of acute antiepileptic drugs did not differ between the groups but long-term antiepileptic drugs were prescribed more often to patients with ASES.

Patients with MACCI also less frequently had favorable functional outcomes (66% vs. 85%, *p* = 0.001), but mortality rates were similar in 2023 (Table 1).

We next analyzed factors associated with outcomes dichotomized into favorable (mRS ≤ 2) and unfavorable (mRS > 2; Table 2) in order to see whether MACCI modifies outcomes in patients with a suspected embolic stroke. Data for the 3-month follow-up were available for all the included patients. In the univariate analysis, MACCI was observed in 61% of the patients with a poor outcome, but in only 35% of the patients that had favorable outcomes (*p* = 0.001). Other factors that were significantly associated with unfavorable outcomes in the univariate analysis included diabetes (58% vs. 40%, *p* = 0.015) and malignancy (29% vs. 10%, *p* < 0.001). In contrast, other vascular risk factors including hypertension, atrial fibrillation, hyperlipidemia, smoking, heart disease, renal failure and prior stroke rates did not significantly differ between the groups. Previous antiplatelet use was more common among patients with unfavorable outcomes (58% vs. 41%, *p* = 0.020) but treatment with any type of anticoagulant prior to the stroke did not differ between the groups.

Clinical presentation with focal signs (68% vs. 48%, *p* = 0.006), encephalopathy (26% vs. 11%, *p* = 0.005) or an altered cognitive state (19% vs. 5%, *p* = 0.001) were also associated with poor outcomes as was the presence of seizures (19% vs. 10%, *p* = 0.048). The presence of pure motor symptoms or the epileptic status upon presentation was not associated with the outcome. ICU admission and increasing stroke severity were also associated with higher odds of unfavorable outcomes (Table 2), whereas treatment with systemic thrombolysis was similar in both groups.

In the multivariable analysis for outcomes (Table 3), MACCI was associated with lower chances of a favorable outcome (OR: 0.190, 95% CI: 0.072–0.502). Similarly, diabetes (OR: 0.222, 95% CI: 0.085 to 0.578), malignancies (OR: 0.153, 95% CI: 0.050 to 0.470) and increasing stroke severity (OR: 0.76, 95% CI: 0.695 to 0.83) were all associated with lower chances for favorable outcomes (Table 3). In contrast, the presence of an altered mental status or seizures at presentation were not associated with outcomes in this analysis.

Among patients with MACCI, five died of pneumonia during hospitalization, two died of severe gastrointestinal bleeding and one died of an intracerebral hemorrhage after systemic thrombolysis with tPA. Compared to survivors, fatalities (Table 4) had higher likelihoods of an altered mental status (31% vs. 7%, *p* = 0.003), higher rates of the epileptic status (15% vs. 1%, *p* = 0.001) and were less often treated with AED (23% vs. 3%, *p* = 0.001). Stroke severity and other baseline criteria did not differ between survivors and fatalities. None of these markers remained significantly associated with mortality in the multivariable analysis (Table 5).

## 4. Discussion

In this study, we compared the clinical manifestations of MACCI patients with those seen in patients with ASES. The main finding of the current study is that MACCI is less often associated with favorable outcomes compared with ASES. Furthermore, we were able to delineate several significant differences between MACCI and ASES. These include higher rates of an altered mental status and seizures upon presentation in MACCI patients. In the current study, the presence of MACCI was confirmed using a short MRI stroke protocol that only included DWI, ADC and FLAIR sequences and could rapidly be achieved for most patients. Our findings suggest that the gold standard for diagnosing MACCI is DWI MRI, which can easily distinguish acute from old ischemic lesions [24,25,26], and is much more sensitive to smaller foci of cortical ischemia compared to non-contrast CT and even CT perfusion studies. Therefore, the use of CT-based screening in these patients could likely lead to significant delays in diagnosing MACCI, which in turn may lead to delays in treatment initiation and prevention of further damage. The current findings may suggest that for patients presenting with atypical symptoms such as an altered mental status or seizures, especially when combined with focal neurological deficits, MRI obtained early during the course of admission may be the optimal screening tool to confirm or refute the diagnosis of a stroke. In this setting, the MRI results can be used as a prognostic tool because MACCI is significantly and independently associated with lower chances for a favorable outcome. While this result may be intuitive since patients with MACCI have involvement of more vascular territories, this is especially compelling since the effect of MACCI on the outcome appears to be independent of initial stroke severity, which was mild in both groups.

Other factors that are associated with the outcome in the current study include initial stroke severity, diabetes and cancer, which are all well-known predictors of the outcome. Interestingly, the presence of seizures or the epileptic status at presentation was not associated with the outcome in the entire cohort and the epileptic status was uncommon in both groups, suggesting that most observed seizures were classified as acute symptomatic seizures in the setting of acute cortical infarctions and did not necessitate long-term treatment. Furthermore, presentation with an altered mental state, seizures or focal neurological signs was not found to influence the chances for favorable outcomes in multivariate analyses after controlling for stroke severity and risk factors. This would suggest that most of these impairments were short-lived and could have resulted from subclinical seizures.

MACCI was identified in 2% of our entire stroke cohort, a figure that is lower than the 10% previously found in other studies [2,3,4,27]. This discrepancy could be attributed to the stricter definition of MACCI adopted in this current study in comparison with previous studies using a more permissive diagnosis [4,5,6]. For example, patients with infarctions in the right ACA and MCA were diagnosed with MACCI in some previous studies but would not have qualified as having MACCI in the current study because both are in the distribution of the right internal carotid. Moreover, the discrepancy could be attributed to the fact that in other published cohorts, all the patients underwent cerebral MRI within the first hours of admission without selection criteria [2,3]. In contrast, DWI was not used routinely as a screening test for all patients with a stroke in our sample and therefore our findings may actually underestimate the prevalence of MACCI. Rather, the acute onset of an altered mental state and unexpected seizures triggered the use of DWI in many of our MACCI patients. Thus, our results suggest the routine use of DWI as a screening test in stroke patients in general, and in patients presenting with an atypical acute altered mental state or unprovoked seizures in particular [27].

Most of the previously published data on MACCI focused on the etiology and imaging features, but little is known about the unique clinical features, comorbidities, stroke characteristics and outcomes of this condition [7]. Furthermore, most of these studies have used different methods of detection of embolic strokes and the data were collected retrospectively with small sample sizes [2,28]. In addition, prior studies focused more on the short-term outcome and in-hospital complications of MACCI patients compared to stroke patients with a single vascular territory involved [29], while some others focused more on the etiology of MACCI [4]. None of the previous studies mentioned has investigated the atypical clinical symptoms and signs upon presentation of patients with MACCI including an altered mental status, encephalopathy and seizures.

Interestingly, patients with MACCI more often had diabetes but less often had ischemic heart disease. The differences in the prevalence of diabetes can possibly be attributed to the older median age of MACCI patients, and the differences in the frequency of ischemic heart disease may suggest a lower prevalence of atherosclerotic disease among patients with MACCI and a higher prevalence of other potential causes of MACCI such as cancer and genetic causes of hypercoagulability or vasculitis. However, the reasons for these differences between the groups are not clear and we cannot totally exclude the possibility of these being chance findings.

The main strengths of the current study include the prospective inclusion of patients into the dataset and the relatively large sample size of MACCI patients.

The main limitations of this study are its retrospective nature and its single-center setting, which may have led to sampling errors and selection bias. Another limitation could be attributed to the fact that 3.0 T MRI was not performed for every patient of our study cohort and some of the patients underwent 1.5 T MRI. More patients with MACCI and smaller less distinct ischemic foci could potentially be identified using the 3.0 T MRI machine [30]. Moreover, data on total infarct volumes were not assessed in this cohort. The generalization of our results to other geographical areas should be explored in future studies.

## 5. Conclusions

Our findings reveal important differences in clinical presentations, associated comorbidities and outcome parameters between MACCI and ASES. Our results suggest that MACCI is an independent predictor of unfavorable outcomes and may represent a more severe form of a stroke despite similar initial stroke severity. Furthermore, our results are of clinical relevance, highlighting the need to perform MRI screening routinely for early identification of MACCI patients presenting with an acutely altered mental state or seizures, avoiding a misdiagnosis and providing guidance for appropriate therapy including earlier initiation of anticoagulation and early management of coagulopathies and cancer in patients with MACCI secondary to these etiologies.

## Figures and Tables

**Figure 1 jcm-12-03973-f001:**
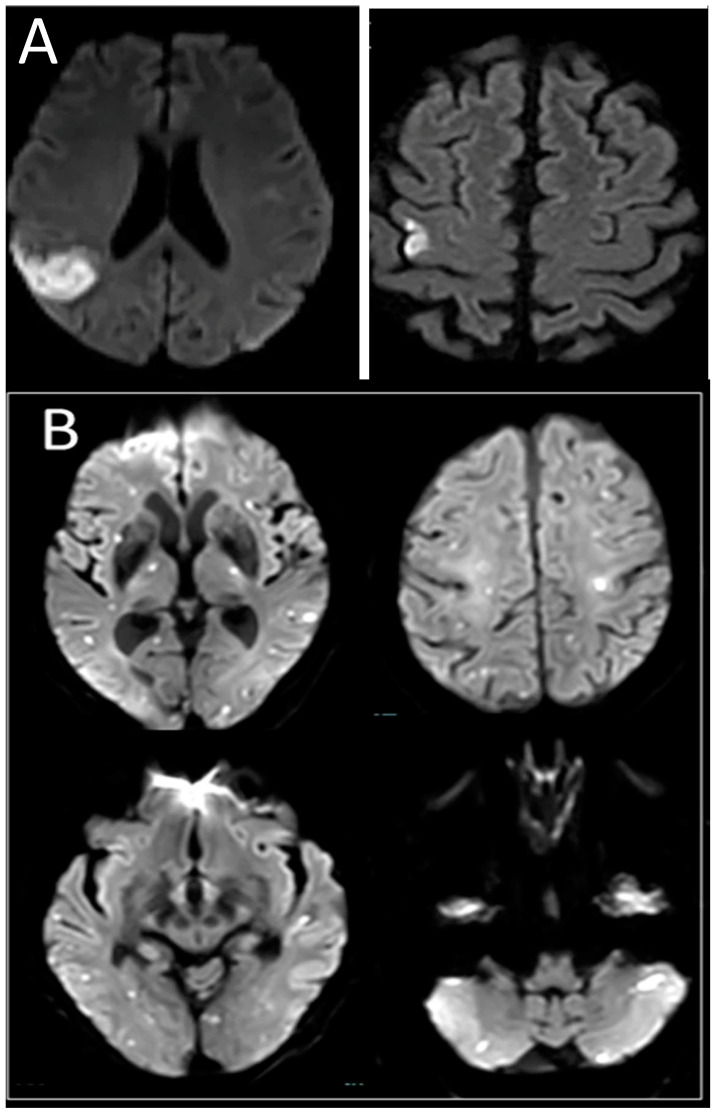
MRI-DWI images from representative patients exhibiting ASES and MACCI. (**A**) ASES showing acute single embolic stroke in the territory of one vessel (right middle cerebral artery). (**B**) MACCI showing multiple embolic stroke in multiple vascular territories.

**Table 1 jcm-12-03973-t001:** Comparison of patients with acute single or multiple concomitant cerebral infarcts.

Characteristics	Single (n = 150)	MACCI (n = 103)	*p*-Value
Age, median (IQR)	68 (58.7–77)	72 (62–79)	0.010
Gender, male (%)	85 (57)	53 (52)	0.461
Comorbidities and risk factors			
Hypertension (%)	105 (70)	79 (77)	0.240
Atrial fibrillation (%)	22 (15)	20 (19)	0.318
Diabetes (%)	56 (37)	55 (53)	0.011
Cholesterol (%)	74 (49)	56 (54)	0.431
Smoking (%)	44 (29)	24 (23)	0.288
Congestive heart failure (%)	36 (24)	19 (18)	0.293
Ischemic heart disease (%)	66 (44)	27 (26)	0.004
Chronic renal failure (%)	15 (10)	14 (14)	0.378
Prior stroke (%)	24 (16)	24 (23)	0.146
Valve (%)	12 (8)	11 (11)	0.487
Antiplatelets (%)	64 (43)	48 (47)	0.491
Coumadin (%)	2 (1)	5 (5)	0.093
NOAC (%)	1 (1)	5 (5)	0.097
NIHSS on admission, median (IQR)	4 (2–8)	4 (2–7)	0.380
Motor impairment (%)	115 (77)	67 (68)	0.117
Focal signs (%)	58 (39)	74 (74)	<0.001
Encephalopathy (%)	9 (6)	28 (27)	<0.001
Cognitive impairment (%)	5 (3)	16 (16)	0.001
Seizures (%)	13 (9)	18 (18)	0.036
Status epilepticus (%)	4 (3)	1 (1)	0.341
Acute AED (%)	6 (4)	5 (5)	0.743
Chronic AED (%)	7 (5)	0 (0)	0.026
Malignancy (%)	17 (12)	18 (18)	0.168
Treatment			
tPA (%)	17 (11)	14 (14)	0.570
ICU (%)	68 (45)	23 (23)	<0.001
Outcomes			
Favorable outcome (%)	128 (85)	68 (66)	<0.001
Mortality (%)	5 (3)	8 (8)	0.117

Values represent the number of patients and their percentages are presented in parentheses unless otherwise stated. IQR—interquartile range, NOAC—novel oral anticoagulants, NIHSS—NIH Stroke Scale/Score, AED—anti-epileptic drugs, tPA—tissue plasminogen activator, ICU—intensive care unit.

**Table 2 jcm-12-03973-t002:** Outcomes in the study population.

Characteristics	Favorable Outcome (n = 196)	Unfavorable Outcome (n = 57)	*p*-Value
Age, median (IQR)	68 (58.3–77)	72 (67–77)	0.094
Gender, male (%)	111 (57)	27 (47)	0.202
Comorbidities and risk factors			
Hypertension (%)	139 (71)	45 (79)	0.231
Atrial fibrillation (%)	30 (15)	12 (21)	0.305
Diabetes (%)	78 (40)	33 (58)	0.015
Cholesterol (%)	95 (49)	35 (61)	0.085
Smoking (%)	58 (30)	10 (18)	0.071
Congestive heart failure (%)	39 (20)	16 (28)	0.188
Ischemic heart disease (%)	70 (36)	23 (40)	0.523
Chronic renal failure (%)	19 (10)	10 (18)	0.102
Prior stroke (%)	33 (17)	15 (26)	0.108
Valve (%)	15 (8)	8 (14)	0.147
Antiplatelets (%)	79 (41)	33 (58)	0.020
Coumadin (%)	6 (3)	1 (2)	0.596
NOAC (%)	7 (4)	2 (4)	0.982
NIHSS on admission, median (IQR)	3 (2–6)	11 (5–18)	<0.001
Motor impairment (%)	138 (72)	44 (79)	0.294
Focal signs (%)	94 (48)	38 (69)	0.006
Encephalopathy (%)	22 (11)	15 (26)	0.005
Cognitive impairment (%)	10 (5)	11 (19)	0.001
Seizures (%)	19 (10)	11 (19)	0.048
Status epilepticus (%)	3 (2)	2 (4)	0.345
Acute AED (%)	7 (4)	4 (7)	0.261
Chronic AED (%)	5 (3)	2 (4)	0.698
Malignancy (%)	19 (10)	16 (29)	<0.001
MACCI (%)	68 (35)	35 (61)	<0.001
Treatment			
tPA (%)	22 (11)	9 (16)	0.362
ICU (%)	63 (32)	28 (49)	0.020

Values represent the number of patients and their percentages are presented in parentheses unless otherwise stated. IQR—interquartile range, NOAC—novel oral anticoagulants, NIHSS—NIH Stroke Scale/Score, AED—anti-epileptic drugs, tPA—tissue plasminogen activator, ICU—intensive care unit, MACCI—multiple acute concomitant infarctions.

**Table 3 jcm-12-03973-t003:** Multivariate analysis for favorable outcome.

	OR	95% CI		*p*-Value
Diabetes	0.207	0.077	0.558	0.002
NIHSS on admission	0.748	0.684	0.817	<0.001
Focal signs	0.222	0.077	0.642	0.005
Encephalopathy	2.598	0.673	10.024	0.166
Cognitive impairment	0.392	0.075	2.038	0.266
Seizures	0.814	0.168	3.935	0.798
Malignancy	0.200	0.063	0.634	0.006
Multiple acute concomitant cerebral infarctions	0.308	0.115	0.827	<0.001

NIHSS—NIH Stroke Scale/Score.

**Table 4 jcm-12-03973-t004:** Survival in the study population.

Characteristics	Survival (n = 240)	Mortality (n = 13)	*p*-Value
Age, median (IQR)	69 (60–77)	70 (63.5–77.5)	0.520
Gender, male (%)	132 (55)	6 (46)	0.522
Hypertension (%)	176 (73)	8 (62)	0.352
Atrial fibrillation (%)	42 (18)	0 (0)	0.099
Diabetes (%)	109 (45)	2 (15)	0.034
Cholesterol (%)	125 (52)	5 (39)	0.339
Smoking (%)	66 (28)	2 (15)	0.337
Congestive heart failure (%)	51 (21)	4 (31)	0.418
Ischemic heart disease (%)	89 (37)	4 (31)	0.646
Chronic renal failure (%)	26 (11)	3 (23)	0.177
Prior stroke (%)	43 (18)	5 (39)	0.066
Valve (%)	22 (9)	1 (8)	0.850
Antiplatelets (%)	108 (45)	4 (31)	0.308
Coumadin (%)	7 (3)	0 (0)	0.532
NOAC (%)	9 (4)	0 (0)	0.477
NIHSS on admission, median (IQR)	4 (2–7)	6.5 (3.25–21)	0.205
Motor impairment (%)	172 (73)	10 (77)	0.749
Focal signs (%)	123 (52)	9 (75)	0.114
Encephalopathy (%)	33 (14)	4 (31)	0.091
Cognitive impairment (%)	17 (7)	4 (31)	0.003
tPA (%)	29 (12)	2 (15)	0.728
ICU (%)	86 (36)	5 (39)	0.856
Seizures (%)	27 (11)	3 (23)	0.199
Status epilepticus (%)	3 (1)	2 (15)	<0.001
Acute AED (%)	8 (3)	3 (23)	0.001
Chronic AED (%)	6 (2)	1 (8)	0.266
Malignancy (%)	31 (13)	4 (33)	0.048
Multiple acute concomitant cerebral infarctions (%)	95 (40)	8 (62)	0.117

Values represent the number of patients and their percentages are presented in parentheses unless otherwise stated. IQR—interquartile range, NOAC—novel oral anticoagulants, NIHSS—NIH Stroke Scale/Score, AED—anti-epileptic drugs, tPA—tissue plasminogen activator, ICU—intensive care unit.

**Table 5 jcm-12-03973-t005:** Multivariate analysis for mortality.

	OR	95% CI		*p*-Value
Diabetes	0.258	0.054	1.233	0.090
Cognitive impairment	3.993	0.999	15.953	0.050
Status epilepticus	1.927	0.099	37.566	0.665
Acute AED	4.367	0.454	42.013	0.202

AED—anti-epileptic drugs.

## Data Availability

Full data are available following a formal request and in compliance with state regulations.

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
