# Peer review of "Characteristics of Multiple Acute Concomitant Cerebral Infarcts Involving Different Arterial Territories"

_jcm, 2023, doi:10.3390/jcm12123973_

Round 1

Reviewer 1 Report

I read with great interest the manuscript by Simaan et al. on the characteristics of Multiple Acute Concomitant Cerebral Infarcts involving different arterial territories. The paper is sound, original, and very well written. However, I have some comments to add.

When you state that " The main etiologies include brain embolism from the heart or the aortic arch, hypoperfusion, hyper-viscosity and coagulopathies", you should also mention that these patients require a focused and frequent assessment of tissue perfusion by the dosage of blood lactate (doi: 10.1155/2022/1192902) and capillary refill time (doi: 10.1186/s12871-022-01920-1), blood coagulation monitoring (doi: 10.1002/ajh.23599) and adequate anti-coagulation therapy (doi: 10.1111/aor.14276). Please discuss and add these references.

Please specify which hospital center the study was conducted in.

You did not mention delirium among the outcomes studied, although it is a frequent manifestation in stroke patients, especially in ICU (doi: 10.3390/jcm12020435). Did you consider it as "cognitive impairment" outcome? Please explain and add this reference.

Author Response

We thank the editor and the reviewers for their thoughtful comments which enabled us to revise our manuscript to be more precise and include all necessary details. To this end, all the points raised have been revised accordingly. As suggested by the editor, a graphical abstract summarizing our work has been added.  

Reviewer #1

1.When you state that " The main etiologies include brain embolism from the heart or the aortic arch, hypoperfusion, hyper-viscosity and coagulopathies", you should also mention that these patients require a focused and frequent assessment of tissue perfusion by the dosage of blood lactate (doi: 10.1155/2022/1192902) and capillary refill time (doi: 10.1186/s12871-022-01920-1), blood coagulation monitoring (doi: 10.1002/ajh.23599) and adequate anti-coagulation therapy (doi: 10.1111/aor.14276). Please discuss and add these references.

We kindly thank the reviewer for this comment. All the cited references are relevant to our research. The introduction has been completely revised and expanded including the relevant references. Please see the new introduction section (page 1-2, lines 43-66).

  1. Please discuss the main etiologies and add the references

MACCI main etiologies have been developed and the appropriate references have been added (page 2, lines 56-66).

  1. Please specify which hospital center the study was conducted in.

It has been specified that the study has been conducted in Hadassah-Hebrew University Medical Center in Material and methods section (page 2, line 72-73).

  1. You did not mention delirium among the outcomes studied although it is a frequent manifestation in stroke patients especially in ICU.

Delirium explanation and appropriate references have been added (page 3, lines 103-107).

Reviewer 2 Report

Authors discuss the characteristics of multiple acute concomitant cerebral infarcts involving different arterial territories. Patients with MACCI were identified from a prospective registry of stroke patients admitted to a tertiary teaching center. MACCI was diagnosed in 103 patients that were compared to 150 patients with ASES. Outcomes are interesting. However, there are some issues that will be addressed. 

- In the introduction, the comparison with previous works must be more precise in order to highlight the real contribution of this work. In addition, the motivation and background of wide practical use of the theoretic results presented should be clearly emphasized to facilitate the readers. In addition, the current introduction is too short to introduce a research paper, it should be rewritten by adding more references. 

- Discussion part is very abstracted, it should be more extended. 

- The provided results are good. However, authors should discuss the limitations. 

- Results are well presented, however, the used method is too reduced and should be more detailed.

- Authors should add the limitations of this work in the conclusion. 

Concluding, the paper has potential to be appreciated by the readers and the above comment are formulated such that to enhance its impact.

English is generally good, but needs to be polished further. The manuscript should be formatted better and some spelling and grammar should be checked carefully.

Author Response

We thank the editor and the reviewers for their thoughtful comments which enabled us to revise our manuscript to be more precise and include all necessary details. To this end, all the points raised have been revised accordingly. As suggested by the editor, a graphical abstract summarizing our work has been added.  

Reviewer 

  1. In the introduction, the comparison with previous works must be more precise in

order to highlight the real contribution of this work. In addition, the motivation and background of wide practical use of the theoretic results presented should be clearly emphasized to facilitate the readers. In addition, the current introduction is too short to introduce a research paper, it should be rewritten by adding more references. 

The introduction has been completely revised and expanded according to the reviewer requirements (page 1-2, lines 43-66).

  1. Discussion part is very abstracted; it should be more extended.

Discussion section has been expanded (page 9, 244-245; page 9, lines 254-256; page 9, lines 266-267; page 9, lines 276-279; page 9lines 282-284; page 10, lines 291-300)

  1. The provided results are good. However, authors should discuss the limitations. 

The limitations have been discussed (page 10, lines 303-309)

  1. Results are well presented, however, the used method is too reduced and should be more detailed.

The methods have been explained in details according to the reviewer requirements (page -3, lines 96-110; page 4, lines 140-152).

  1. authors should add the limitation of his work un the conclusion.

The limitations have been added to the discussion (page 10, lines 303-309)

Round 2

Reviewer 2 Report

The authors reacted properly to my pointed issues.

Author Response

We thank the editor and the reviewers.
we have already responded to the reviewer's comments.
please highlight any other questions and we will gladly respond .